# L-FMLC: End-to-End Neuro-Fuzzy Learning for Adaptive and Scalable Interpretability

## Abstract

Deep learning models offer remarkable performance but their black-box nature remains a key barrier to deployment in high-stakes domains. Neuro-fuzzy systems provide a promising path towards inherent interpretability but often struggle with adaptability and scalability. Our prior work, FMLC, coupled a deep model with a Type-I TSK-style output layer but relied on static, pre-defined fuzzy sets and suffered from rule explosion on complex problems. This paper introduces L-FMLC, a framework with a fully adaptive fuzzification layer that addresses these limitations. The layer learns the optimal positions and shapes of its membership functions and autonomously determines the required number of fuzzy sets for each feature via a novel, regularized gating mechanism with an explicit pruning policy. This structural learning is guided by interpretability-driven regularizers that ensure the resulting fuzzy partitions are parsimonious and semantically coherent. To manage the subsequent rule base, we introduce a two-stage distillation framework that combines architectural pruning with hierarchical clustering to condense thousands of rules into a compact, high-fidelity set of "meta-rules". We demonstrate that these structured rules can be translated into natural language explanations using a large language model. Experiments against strong interpretable baselines like EBM and XGBoost show that our regularized L-FMLC framework achieves competitive or superior predictive accuracy. Crucially, ablation studies provide compelling empirical evidence that this regularization is essential not just for interpretability, but also for generalization.

## 1 Introduction

The proliferation of large-scale deep learning models has revolutionized AI, yet this success has created a critical tension: as models grow more powerful, they often become more opaque. This lack of transparency hinders their adoption in high-stakes domains like finance, healthcare, and science, where trust and accountability are non-negotiable Barredo Arrieta et al. (2020). In response, the field of Explainable AI (XAI) has emerged, with a key branch focusing on building *inherently interpretable* or "glass-box" models.

Neuro-fuzzy systems, particularly those inspired by the Takagi-Sugeno-Kang (TSK) model Takagi & Sugeno (1985), represent a powerful class of such glass-box models. They are designed to combine the potent learning capabilities of neural networks with the transparent, human-readable rule-based structure of fuzzy logic (e.g., "IF... THEN...").

In prior work Anonymous (2025), we introduced the Fuzzy-Modulated Linear Consequents (FMLC) framework, which uses a deep model to dynamically generate the coefficients of a final linear layer. Despite its promise, this initial framework was constrained by two fundamental limitations: 1) Its reliance on static, task-agnostic fuzzification, and 2) its susceptibility to a combinatorial rule explosion on high-dimensional data, undermining practical interpretability.

To address these challenges, this paper introduces **Learnable FMLC (L-FMLC)**, a substantially advanced framework that enables true end-to-end learning and scalable interpretability. Our key contributions are:

- **A Regularized Adaptive Fuzzification Layer.** We propose a novel architecture that learns not only the parameters of its membership functions but also autonomously determines the

optimal *number* of fuzzy sets for each feature. This is achieved through a learnable gating mechanism, guided by sparsity, overlap, and coverage regularizers, and coupled with an explicit pruning policy to ensure structural parsimony.

- **A Scalable, Two-Stage Interpretability Framework.** We address the rule explosion problem by first leveraging the architectural pruning from our adaptive layer, and then applying hierarchical clustering to distill thousands of raw rules into a compact set of quantitative "meta-rules". We demonstrate that these structured outputs can serve as effective prompts for LLMs to generate high-level, natural language explanations and quantify their fidelity.

- **Empirical Evidence for Regularization-Driven Performance.** Through rigorous experiments against strong baselines (e.g., EBM, XGBoost) and ablation studies, we show that L-FMLC achieves competitive performance. We provide compelling evidence that our interpretability-guided regularization is essential not just for obtaining coherent rules, but for unlocking strong predictive accuracy by guiding the highly flexible architecture towards generalizable solutions.

## 2 RELATED WORK

**Neuro-Fuzzy Systems and Learnable Parameters.** The integration of neural networks and fuzzy logic has a rich history, from shallow models like ANFIS Jang (1993) to deeper architectures Angelov & Soares (2020). A central challenge in making membership function (MF) parameters learnable is that unconstrained optimization can lead to semantically meaningless solutions (e.g., collapsed or redundant MFs). L-FMLC directly addresses this gap by coupling end-to-end learning with novel, interpretability-guided regularization, ensuring that adaptivity does not come at the cost of comprehensibility.

**Scalable and Accessible Model Interpretability.** A central challenge for any interpretable model is scalability. While *post-hoc* methods like LIME Ribeiro et al. (2016) and SHAP Lundberg & Lee (2017) explain black-box models, inherently interpretable models often face their own challenges. Rule-based systems suffer from the "curse of dimensionality", while models like Generalized Additive Models (GAMs) can struggle with concurvity Siems et al. (2023). Our work is part of a growing effort to build scalable "white-box" models. Competitors include tree-based ensembles like XGBoost Chen & Guestrin (2016) and fully interpretable models like Explainable Boosting Machines (EBMs) Nori et al. (2019), which are based on GAMs. Our hierarchical rule distillation connects to the idea of making explanations accessible, recently explored by leveraging LLMs to create narratives from structured data Martens et al. (2024). L-FMLC contributes a method that is both intrinsically interpretable at the local level and scalable to a global, high-level understanding.

**Dynamic Parameter Estimation.** The core mechanism of L-FMLC, where a `DeepModel` generates modulators for another layer, is conceptually related to hypernetworks Ha et al. (2017) and attention Vaswani et al. (2023). While sharing this meta-learning DNA, L-FMLC's purpose is distinct and tailored for interpretability. The modulators are explicitly generated from *fuzzified inputs*, and their sole function is to parameterize a final Type-I TSK-style *linear layer* that operates on the original features. This unique structure is what enables the extraction of dynamic linear equations and fuzzy rules, differentiating it from general-purpose approaches.

## 3 THE L-FMLC FRAMEWORK

L-FMLC is a comprehensive neuro-fuzzy framework designed for high performance and multi-level interpretability. It introduces two critical innovations to our prior work Anonymous (2025): an end-to-end adaptive fuzzification layer and a scalable rule distillation module.

### 3.1 PRELIMINARIES AND NOTATION

Let the input be a feature vector $X \in \mathbb{R}^S$. Our model uses a set of fuzzy membership functions (MFs) to partition each feature's domain. We start with a maximum of $K_{\max}$ potential MFs per feature. The parameters for the $k$-th MF of the $s$-th feature are its center $c_{s,k}$ and standard deviation

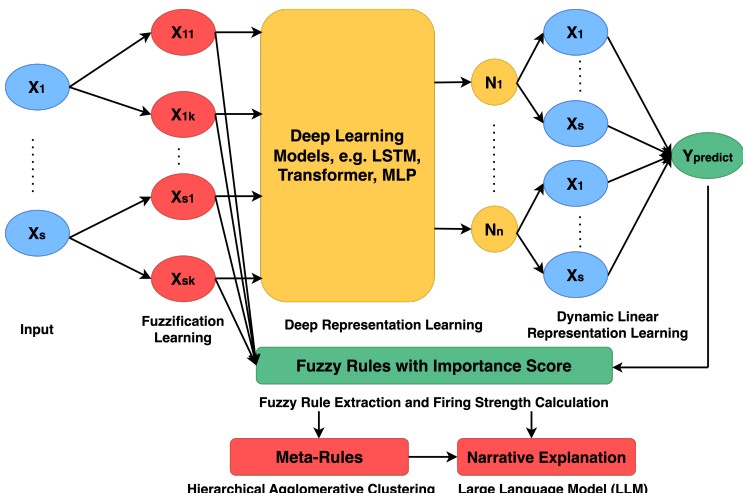

Figure 1: The L-FMLC framework. The **prediction path (top)** uses a deep model on adaptive fuzzified inputs to generate modulators ($N$) for the final linear layer. The **interpretation pipeline (bottom)** extracts Type-I TSK rules, distills them into "meta-rules" via clustering, and finally leverages an LLM to generate human-readable explanations.

$\sigma_{s,k}$. A learnable gate, $g_{s,k} \in [0, 1]$, controls the activity of each MF. The model's prediction, $y_p$, is a linear combination of the input features, where the coefficients are dynamically generated by a `DeepModel` that processes the fuzzified inputs.

## 3.2 CORE MECHANISM: DYNAMIC CONSEQUENTS

L-FMLC inherits its core predictive mechanism from FMLC Anonymous (2025). Given an input $X$, it is first mapped to a fuzzified representation $M \in [0, 1]^{S \times K}$ (details in Sec. 3.3). A `DeepModel` (e.g., a Transformer or LSTM) processes $M$ to produce a vector of $n_N$ context **modulators**, $N \in \mathbb{R}^{n_N}$. The final prediction is a dynamically weighted linear combination of original features:

$$y_p = \sum_{s=1}^{S} \left( \sum_{i=1}^{n_N} W_{s,i} \cdot N_i \right) \cdot X_s \tag{1}$$

where $W \in \mathbb{R}^{S \times n_N}$ is a matrix of trainable weights. This provides a direct, instance-specific linear explanation. A constant feature $X_0 = 1$ can be appended to the input to learn a dynamic bias term.

## 3.3 INNOVATION 1: ADAPTIVE FUZZIFICATION

The first key innovation is a fully adaptive fuzzification layer that learns the optimal shape, position, and *number* of fuzzy sets for each feature.

**Differentiable and Gated Membership Functions.** We use a Gaussian MF for its smoothness (Eq. 2), where $c_{s,k}$ and $\sigma_{s,k}$ are trainable.

$$\mu_{s,k}(x_s; c_{s,k}, \sigma_{s,k}) = \exp\left(-\frac{(x_s - c_{s,k})^2}{2\sigma_{s,k}^2}\right) \tag{2}$$

To learn the number of sets, we introduce a learnable gate $g_{s,k}$ for each of the $K_{\max}$ potential MFs. To maintain the interpretability of a membership degree, we constrain the gates to $[0, 1]$ by parameterizing them with a sigmoid function, $g_{s,k} = \text{sigmoid}(\alpha_{s,k})$, where $\alpha_{s,k}$ are the learnable logits initialized to a value that makes $g_{s,k} \approx 1$. The final gated membership degree $\hat{\mu}_{s,k}$ is:

$$\hat{\mu}_{s,k}(x_s) = g_{s,k} \cdot \mu_{s,k}(x_s; c_{s,k}, \sigma_{s,k}) \tag{3}$$

The vector of all $\hat{\mu}_{s,k}(x_s)$ values for a given sample forms the fuzzified representation $M$.

**Regularization and Pruning Policy.** Unconstrained learning can yield semantically poor MFs. We guide the process with three regularization terms and an explicit pruning policy.

- **Sparsity Regularization** ($L_{\textbf{sparse}}$)**:** We apply an L1 penalty to the gate logits $\alpha_{s,k}$ to encourage gates to move towards 0.

$$L_{\text{sparse}} = \sum_{s,k} |\alpha_{s,k}| \tag{4}$$

  While L1 regularization encourages sparsity, it does not guarantee exact zeros during gradient-based optimization. We therefore introduce a hard pruning step.

- **Explicit Pruning Policy:** Post-training, we define a small threshold $\tau$ (e.g., $10^{-3}$). Any MF whose gate $g_{s,k}$ is below $\tau$ is considered inactive and is permanently pruned from the model architecture for inference and rule extraction. This makes the structural learning decisive. More advanced techniques like hard-concrete gates Louizos et al. (2018) could also be used.

- **Overlap Regularization** ($L_{\textbf{overlap}}$)**:** This term penalizes adjacent MFs from becoming too similar. Let $\mathcal{A}_s = \{k \mid g_{s,k} > \tau\}$ be the set of active MF indices for feature $s$. Let the centers of these active MFs, sorted in ascending order, be $c'_{s,j}$. The regularizer is applied to consecutive pairs in this sorted list:

$$L_{\text{overlap}} = \sum_s \sum_{j=1}^{|\mathcal{A}_s|-1} \exp\left(-\frac{(c'_{s,j+1} - c'_{s,j})^2}{(\sigma'_{s,j})^2 + (\sigma'_{s,j+1})^2}\right) \tag{5}$$

- **Coverage Regularization** ($L_{\textbf{coverage}}$)**:** This term encourages the active MFs to span the observed data range, preventing clustering.

$$L_{\text{coverage}} = \sum_s \left(\left(\min_{k \in \mathcal{A}_s} c_{s,k} - x_{s,\min}\right)^2 \right.$$
$$\left. + \left(\max_{k \in \mathcal{A}_s} c_{s,k} - x_{s,\max}\right)^2\right) \tag{6}$$

  While using hard min/max is non-smooth and sensitive to outliers, in practice it works well with Adam. For greater stability, one could use robust quantiles (e.g., 5th/95th percentiles) instead of absolute min/max.

The total loss is a weighted sum of the task loss and these regularizers: $L_{\text{total}} = L_{\text{task}} + \lambda_{\text{sp}}L_{\text{sparse}} + \lambda_{\text{ov}}L_{\text{overlap}} + \lambda_{\text{cov}}L_{\text{coverage}}$.

### 3.4 INNOVATION 2: SCALABLE RULE DISTILLATION

The second innovation manages the "rule explosion" problem.

**Formalizing Rule Extraction.** A Type-I TSK fuzzy rule is defined by an antecedent (a fuzzy region in the input space) and a consequent (a linear function). An antecedent is a combination of one fuzzy set for each of the $S$ features, e.g., "IF $X_1$ is $\text{MF}_{1,k_1}$ AND ... AND $X_S$ is $\text{MF}_{S,k_S}$". The total number of potential rules is $\prod_s |\mathcal{A}_s|$.

For each training sample, we identify the single most-activated rule antecedent, defined by the set of "winner" fuzzy sets $\{k_s^*\}_{s=1}^S$ where $k_s^* = \arg\max_k \hat{\mu}_{s,k}(x_s)$. We consider only those unique rule antecedents that are activated by at least one training sample. For each such activated rule $P$, we compute its characteristic modulator vector, $N_{P,\text{char}}$, by averaging the modulator vectors $N$ from all training samples that activate $P$. The TSK-style rule is then: IF X activates $Antecedent(P)$, THEN $y_p = \sum_{s=1}^S \left(\sum_{i=1}^{n_N} W_{s,i} \cdot N_{P,\text{char},i}\right) \cdot X_s$.

**Rule Vectorization and Hierarchical Clustering.** To cluster rules, each activated rule $P$ is vectorized by concatenating its antecedent parameters (the centers $c_{s,k_s^*}$ and spreads $\sigma_{s,k_s^*}$ for its winner MFs) and its consequent coefficients ($C_{P,s} = \sum_i W_{s,i}N_{P,\text{char},i}$). After standardizing these vectors, we apply Hierarchical Agglomerative Clustering (HAC) Ward (1963) to group them. The number

---

**Algorithm 1** L-FMLC Rule Distillation Pipeline

---

**Input**: Trained L-FMLC model, training data $X_{train}$

1: Extract all unique activated rule antecedents $\mathcal{P}$ from $X_{train}$ by finding the winner MF for each feature per sample.
2: Initialize empty lists $V_{rules}, C_{rules}$.
3: **for** each rule antecedent $P \in \mathcal{P}$ **do**
4:      Compute $N_{P,\text{char}}$ by averaging modulators $N$ over samples activating $P$.
5:      Compute consequent coefficients $C_P$.
6:      Form antecedent vector $v_{ant}$ from MF params of $P$.
7:      $V_{rules}$.append(concatenate($v_{ant}, C_P$)).
8:      $C_{rules}$.append($C_P$).
9: **end for**
10: Standardize $V_{rules}$.
11: Apply HAC on $V_{rules}$ to get cluster assignments $\mathcal{K}$.
12: Determine optimal number of clusters by maximizing silhouette score.
13: **for** each cluster $k \in \mathcal{K}$ **do**
14:      Find medoid rule $P_k^*$ in cluster $k$.
15:      Meta-antecedent $\leftarrow$ Antecedent($P_k^*$).
16:      $\bar{C}_{\text{meta},k} \leftarrow$ mean($C_P$ for rules in cluster $k$).
17:      Form meta-rule $k$ from Meta-antecedent and $\bar{C}_{\text{meta},k}$.
18: **end for**
19: **return** Meta-rules.

---

of meta-rule clusters is determined automatically by selecting the cut-point on the dendrogram that maximizes the average silhouette score.

For each resulting cluster, we distill a single **meta-rule**:

1. The **meta-antecedent** is from the medoid rule of the cluster (the most representative rule).

2. The **meta-consequent** is a new coefficient vector, $\bar{C}_{\text{meta}}$, calculated by averaging the consequent vectors $C_P$ of all rules within the cluster.

This process, summarized in Algorithm 1, transforms an unmanageable rule base into a handful of interpretable meta-rules.

## 4 THEORETICAL ANALYSIS

The L-FMLC architecture maintains strong expressive power. We extend the universal approximation theorem from prior work Anonymous (2025) to our setting.

**Theorem 1** (Universal Approximation for L-FMLC). *Let an L-FMLC model be defined as in Section 3. Assume its MFs are continuous, its* `DeepModel` *is a universal approximator, and the input domain $D \subset \mathbb{R}^S$ is compact. For any target function $f(X) = \sum_{s=1}^{S} g_s(X) \cdot X_s$, where each coefficient function $g_s(X)$ is continuous, and for any $\epsilon > 0$, there exists an L-FMLC configuration such that $\sup_{X \in D} |f(X) - y_{L\text{-}FMLC}(X)| < \epsilon$.*

**Proof Sketch.** The proof, detailed in the Appendix, leverages the universal approximation capability of the `DeepModel` to show that the generated modulators $N$ can form a basis rich enough to approximate any continuous coefficient functions $g_s(X)$ via the linear combination with weights $W$. The continuity of the learnable fuzzification layer ensures this property holds.

## 5 EXPERIMENTS

We conduct experiments to answer: (1) Does L-FMLC achieve competitive performance against strong interpretable and black-box baselines? (2) Does our regularization produce more meaningful fuzzy partitions and better generalization? (3) Is our rule distillation framework effective and does it maintain fidelity?

### 5.1 EXPERIMENTAL SETUP

**Datasets.** (1) Electricity Transformer Temperature (ETT): A standard time-series benchmark Zhou et al. (2021). We use the ETTh1 dataset (hourly data) to forecast the 'oil temperature' variable 24 hours ahead, using a lookback window of 96 hours ($S = 96 \times 7$ features). We use standard chronological splits. (2) Solar Irradiation Prediction: A high-dimensional regression task ($S = 20$) from the NSRDB Sengupta et al. (2018) to predict Global Horizontal Irradiance. (3) Breast Cancer Wisconsin: A standard binary classification task ($S = 30$) from the UCI Repository Zwitter & Soklic (1988).

**Models for Comparison.** (1) Interpretable Baselines: Explainable Boosting Machine (EBM) Nori et al. (2019), XGBoost Chen & Guestrin (2016), and ARIMA (for time-series). (2) Deep Baselines: LSTM and Transformer models trained on raw inputs. (3) Ablation Models: Static FMLC Anonymous (2025) and our L-FMLC (w/o Reg).

**Implementation Details.** Deep models were implemented in TensorFlow and trained with Adam. All results are reported as **mean ± std. dev.** over 10 runs with different random seeds. For L-FMLC, we set $K_{\max}$ to be intentionally larger than the expected number of MFs and used a pruning threshold $\tau = 10^{-3}$. Full implementation details are in the Appendix.

### 5.2 PREDICTIVE PERFORMANCE

Tables 1 and 2 show the results. The findings highlight a crucial trade-off between pure performance and interpretability.

On the regression tasks (Table 1), our full model, **L-FMLC (w/ Reg)**, is highly competitive. On the Solar dataset, it significantly outperforms all other models, including the powerful XGBoost. On the ETT time-series task, it performs on par with the specialized Transformer model and substantially better than ARIMA and Static FMLC. This demonstrates its effectiveness in capturing complex relationships.

The ablation results are critical. **L-FMLC (w/o Reg)** consistently underperforms, often doing worse than the simpler Static FMLC. This provides strong empirical evidence that our regularization is not merely for interpretability but is *essential for generalization*, guiding the flexible architecture away from overfitting.

On the Breast Cancer classification task (Table 2), EBM and XGBoost show very strong performance, as expected for gradient-boosted trees on clean tabular data. Our L-FMLC (w/ Reg) is highly competitive, achieving a near-perfect AUC-ROC, and significantly outperforming the standard MLP. While EBM may be marginally better in pure accuracy here, L-FMLC offers a different paradigm of interpretability through instance-specific linear equations and global fuzzy rules.

### 5.3 ANALYSIS OF LEARNABLE FUZZIFICATION

Figure 2 provides a powerful visual demonstration of our regularization's impact on the "Total Cloud Cover" feature from the Solar dataset, which has a bimodal distribution. The static FMLC (a) uses a task-agnostic, inefficient partition. The unregularized L-FMLC (b) creates a degenerate, collapsed solution, failing to prune any of the $K_{\max} = 10$ MFs. In contrast, our full L-FMLC (c) successfully prunes 7 redundant MFs, discovering an optimal and semantically meaningful $K = 3$ structure corresponding to "Clear", "Partly Cloudy" and "Overcast" conditions.

### 5.4 ANALYSIS OF SCALABLE RULE MANAGEMENT

On the Solar dataset, Static FMLC with $K = 5$ for 20 features would generate a combinatorially explosive number of potential rules. In contrast, our L-FMLC's adaptive layer first performed **architectural pruning**, learning a non-uniform set of active MFs per feature (e.g., "[3, 4, 2, 5,...]"). This reduced the number of activated rules found in the training set from thousands to just **298**.

Table 1: Predictive Performance on Regression Tasks. Results are mean ± std. dev. over 10 runs. Best interpretable model in **bold**, best overall in underline.

| Dataset | Model | MSE (↓) | MAE (↓) |
|---------|-------|---------|---------|
| ETT (h1) | ARIMA | 0.851 ± 0.00 | 0.692 ± 0.00 |
| | Transformer | 0.512 ± 0.011 | 0.488 ± 0.009 |
| | Static FMLC-Transformer | 0.589 ± 0.015 | 0.541 ± 0.011 |
| | L-FMLC-Transformer (w/o Reg) | 0.615 ± 0.021 | 0.580 ± 0.018 |
| | **L-FMLC-Transformer (w/ Reg)** | **0.519 ± 0.013** | **0.495 ± 0.010** |
| Solar | XGBoost | 8891.5 ± 110.2 | 70.1 ± 1.5 |
| | EBM | 9105.3 ± 95.7 | 71.8 ± 1.1 |
| | Transformer | 8100.1 ± 150.3 | 65.9 ± 1.8 |
| | Static FMLC-Transformer | 9267.9 ± 145.2 | 73.4 ± 1.6 |
| | L-FMLC-Transformer (w/o Reg) | 10011.8 ± 198.4 | 79.2 ± 2.1 |
| | **L-FMLC-Transformer (w/ Reg)** | **8341.1 ± 130.5** | **67.2 ± 1.4** |

Table 2: Predictive Performance on Breast Cancer Classification. Best interpretable in **bold**, best overall in underline.

| Model | AUC-ROC (↑) |
|-------|-------------|
| XGBoost | 0.9985 ± 0.0010 |
| EBM | 0.9981 ± 0.0012 |
| MLP (Standard) | 0.9665 ± 0.0040 |
| Static FMLC-MLP | 0.9975 ± 0.0015 |
| L-FMLC-MLP (w/o Reg) | 0.9943 ± 0.0021 |
| **L-FMLC-MLP (w/ Reg)** | **0.9979 ± 0.0013** |

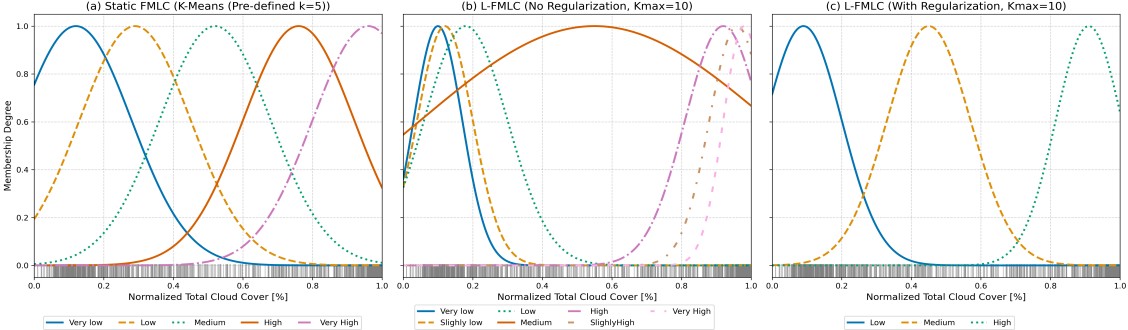

Figure 2: Comparison of fuzzy partitions for the bimodal "Total Cloud Cover [%]" feature. (a) Static FMLC uses a pre-defined, task-agnostic K=5 partition. (b) The unregularized L-FMLC fails to prune redundant sets, creating a degenerate, collapsed solution. (c) Our regularized L-FMLC successfully prunes 7 unnecessary MFs, automatically discovering an optimal and highly interpretable K=3 structure.

Next, HAC was applied to these 298 rules. The dendrogram (Figure 3) was cut to produce 5 meta-clusters based on maximizing the silhouette score. The resulting meta-rules are summarized in Table 3.

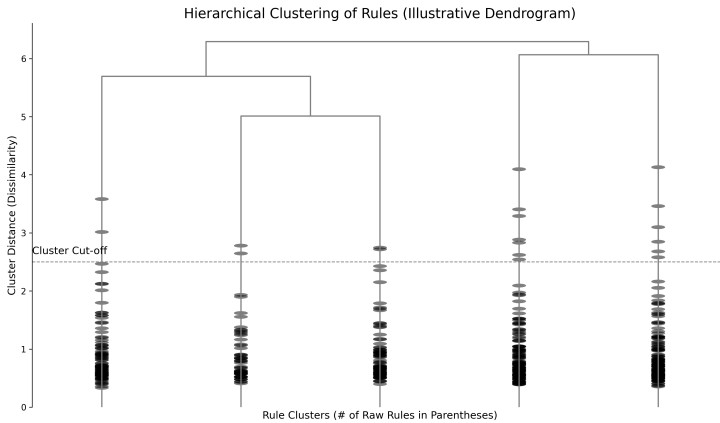

Figure 3: Truncated dendrogram of the 298 TSK rules. The cut-off line, determined by maximizing silhouette score, partitions the rules into 5 distinct meta-clusters.

Table 3: Distilled Meta-Rules for the Solar Irradiation Model.

| Meta-Rule Cluster | Qualitative Description (Dominant Antecedent) | Rules Repr. |
|---|---|---|
| 1. Low Light | IF Solar Zenith Angle is *High* AND Cloud Cover is *Overcast* | **110** |
| 2. Ideal Sunny Day | IF Solar Zenith Angle is *Low* AND Clear Sky Index is *High* | **90** |
| 3. Humid / Overcast | IF Temperature is *High* AND Cloud Cover is *Mostly Overcast* | **55** |
| 4. Cold & Clear Day | IF Solar Zenith Angle is *High* AND Clear Sky Index is *High* | **30** |
| 5. Scattered Clouds | IF Cloud Cover is *Partly Cloudy* AND GHI is *Medium* | **13** |

**Quantitative Evaluation of Meta-Rules.** To validate the rule distillation process, we measure its **fidelity**: how well do the simplified meta-rules approximate the full set of 298 rules? We compute predictions on a held-out test set using both the full rule base (by finding the activated rule for each sample) and the meta-rule base (by finding the activated meta-rule). The Pearson correlation between these two sets of predictions serves as our fidelity score. As shown in Table 4, the meta-rules achieve a fidelity of **0.941**, indicating they capture the vast majority of the original model's logic while being drastically more compact.

Table 4: Quantitative Evaluation of Meta-Rule Distillation.

| Metric | Value |
|---|---|
| Raw Activated Rules | 298 |
| Distilled Meta-Rules | 5 |
| Fidelity (Pearson's r) | 0.941 |

Finally, the structured output of a meta-rule serves as an effective prompt for an LLM to generate a qualitative narrative, as shown in the Appendix, bridging the gap from quantitative logic to human-centric explanation.

## 6 CONCLUSION

In this paper, we introduced L-FMLC, a fully adaptive neuro-fuzzy framework. Its novel, regularized fuzzification layer autonomously learns the optimal number and placement of fuzzy sets. To address rule explosion, we presented a two-stage interpretability framework that combines inherent architectural pruning with high-fidelity hierarchical clustering. Our experiments, conducted against

strong interpretable baselines, confirm that L-FMLC achieves competitive predictive accuracy and establish that our interpretability-guided regularization is essential for guiding the flexible architecture to a generalizable solution. L-FMLC offers a robust pathway towards building AI systems that are simultaneously more adaptive, transparent, and high-performing.

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

# A   APPENDIX

## A.1   UNIVERSAL APPROXIMATION CAPACITY OF L-FMLC

This section provides a more detailed proof for the universal approximation capability of the L-FMLC framework, which was stated in Theorem 1 in the main paper. The theorem states that L-FMLC can approximate any continuous function of the form $f(X) = \sum_{s=1}^{S} g_s(X) \cdot X_s$ on a compact domain, where $g_s(X)$ are continuous coefficient functions.

*Proof Sketch.* The proof relies on demonstrating that the dynamically generated coefficients $C_s(X)$ of the L-FMLC model can arbitrarily approximate any set of continuous target coefficient functions $g_s(X)$. The total approximation error can then be shown to be arbitrarily small. This requires a set of standard assumptions.

**Assumptions:**

**A.1** The input domain $X \in D$ is a compact subset of $\mathbb{R}^S$.

**A.2** The base membership functions $\mu_{s,k}(\cdot)$ (e.g., Gaussian) are continuous. The overall fuzzification map from an input $X$ to the gated membership vector $M(X)$ is continuous, as it is a composition of continuous functions (MFs, gates with sigmoid activation, etc.).

**A.3** The `DeepModel` is a universal approximator for continuous vector-valued functions on a compact domain. This is a standard property for sufficiently large MLPs with appropriate non-linear activations Cybenko (1989).

The proof proceeds constructively:

1. **Approximating Target Coefficients** $g_s(X)$**:** By the Stone-Weierstrass theorem, any continuous function can be approximated by a linear combination of basis functions. The outputs of the `DeepModel`, the modulators $N(M(X))$, serve as these basis functions. By Assumption A.3, the `DeepModel` can be configured to produce a set of modulators $N_i$ whose span is dense in the space of continuous functions on the compact domain of $M(X)$. Therefore, for any set of continuous target coefficient functions $g_s(X)$, there exist weights $\tilde{W}_{s,i}$ such that the ideal dynamic coefficient, $C_s^*(X) = \sum_i \tilde{W}_{s,i} \cdot N_i(M(X))$, can arbitrarily approximate $g_s(X)$. That is, for any $\delta > 0$, we have $|g_s(X) - C_s^*(X)| < \delta$.

2. **Bounding Total Approximation Error:** The total error between the target function $f(X)$ and the L-FMLC output $y_{\text{L-FMLC}}(X)$ is:

$$|f(X) - y_{\text{L-FMLC}}(X)| = \left| \sum_s (g_s(X) - C_s(X)) \cdot X_s \right|$$

$$\leq \sum_s |g_s(X) - C_s(X)| \cdot |X_s|$$

Since $X$ is on a compact domain, $|X_s|$ is bounded by some constant $B$. As the coefficient error $|g_s(X) - C_s(X)|$ can be made arbitrarily small (less than $\delta$), the total error can be bounded: $\sum_s \delta \cdot B$. By choosing a sufficiently expressive `DeepModel` and appropriate weights $W$, this error can be made less than any given $\epsilon > 0$.

This demonstrates that the L-FMLC architecture is sufficiently expressive. The regularization terms guide the optimization towards solutions within this expressive space that are also interpretable, without fundamentally limiting the approximation capacity. $\square$

## A.2 CONVERGENCE ANALYSIS OF L-FMLC TRAINING

We analyze the convergence of training L-FMLC by minimizing the non-convex loss function $L_{\text{total}}$ using stochastic optimization methods like Adam. Proving convergence to a global minimum is intractable. Instead, we show that under standard assumptions, training converges to a first-order stationary point.

**Assumptions:** Let $\Theta$ be the set of all trainable parameters in L-FMLC.

**B.1** The total loss function $L_{\text{total}}(\Theta)$ is L-smooth (its gradient is Lipschitz continuous). This is a reasonable assumption as all components of L-FMLC are smooth functions.

**B.2** The stochastic gradients computed on mini-batches are unbiased estimators of the true gradient.

**B.3** The variance of the stochastic gradients is bounded.

**Convergence Guarantee.** Under these standard assumptions, stochastic optimization theory guarantees that for a non-convex, L-smooth objective, algorithms like SGD with a suitable decaying learning rate will converge in expectation to a stationary point Bottou et al. (2018). That is, $\lim_{T \to \infty} \frac{1}{T} \sum_{t=1}^{T} \mathbb{E}[\|\nabla L_{\text{total}}(\Theta_t)\|^2] = 0$. While Adam, used in our experiments, has more complex dynamics, its convergence in the non-convex setting is also well-studied and points to finding stationary points Reddi et al. (2019). This provides theoretical grounding that our training procedure is stable and finds meaningful solutions.

## A.3 EXPERIMENTAL SETUP AND IMPLEMENTATION DETAILS

This section provides supplementary details on datasets, model configurations, and training protocols to ensure full reproducibility.

### A.3.1 DATASET DETAILS

- **Electricity Transformer Temperature (ETT):** We use the ETTh1 dataset from the public benchmark collection Zhou et al. (2021). The task is long-sequence forecasting: predict

the next 24 hourly values of the 'oil temperature' target variable, given a lookback window of 96 hours of all 7 features. We use the standard 60%/20%/20% chronological split for training, validation, and testing. All features are z-score normalized based on the training set statistics.

- **Solar Irradiation Prediction:** This regression task uses data from the National Solar Radiation Database (NSRDB) Sengupta et al. (2018) with 20 meteorological features ($S = 20$) to predict Global Horizontal Irradiance. The data is split 80%/10%/10% for training, validation, and testing.

- **Breast Cancer Wisconsin (Diagnostic):** This widely-used binary classification dataset from the UCI Repository Zwitter & Soklic (1988) contains 569 samples with 30 features ($S = 30$). The data was randomly split 70%/15%/15% for training, validation, and testing.

### A.3.2 HYPERPARAMETER TUNING AND TRAINING PROTOCOL

Hyperparameters for all models were tuned via grid search on the validation set. All deep learning models were implemented in TensorFlow 2.14 and trained with the Adam optimizer Kingma & Ba (2017) for up to 100 epochs with early stopping based on validation loss.

- **L-FMLC/Static FMLC:** Learning rates were searched in $\{10^{-4}, 5 \cdot 10^{-4}, 10^{-3}\}$, batch sizes in $\{32, 64, 128\}$, and the number of modulators $n_N$ in $\{4, 8, 16\}$. For L-FMLC, regularization weights $\lambda_{sp}, \lambda_{ov}, \lambda_{cov}$ were searched in the range $[10^{-6}, 10^{-2}]$. The gate pruning threshold $\tau$ was set to $10^{-3}$.

- **EBM and XGBoost:** We used the official implementations from 'interpret' and 'xgboost' libraries. Key hyperparameters like 'n_estimators', 'max_depth', 'learning_rate', and 'subsample' were tuned via grid search.

- **Fuzzification Parameters:** The number of fuzzy sets used for Static FMLC ($K_{fix}$) and the maximum number for L-FMLC ($K_{max}$) are detailed in Table 5.

Table 5: Number of Fuzzy Sets (K) Used in Experiments.

| Dataset | $K_{fix}$ (Static FMLC) | $K_{max}$ (L-FMLC variants) |
|---|---|---|
| ETT (h1) | 5 | 10 |
| Solar Irradiation | 5 | 10 |
| Breast Cancer | 3 | 5 |

### A.4 LLM-POWERED NATURAL LANGUAGE VERBALIZATION

As a final step to maximize accessibility, the structured output of a meta-rule can serve as a prompt for a Large Language Model (LLM). Figure 4 shows a case study where we take the "Ideal Sunny Day" meta-rule from the Solar Irradiation task and prompt a generic LLM to translate its quantitative logic into a qualitative narrative. This demonstrates the potential of our framework to generate human-centric explanations from its distilled logical structure.

**Input to LLM (Prompt):**

```
# Explain this TSK fuzzy rule from a solar prediction model
# Context: Predicts solar irradiation (GHI).
# Meta-Rule: "Ideal Sunny Day"
# ANTECEDENT (Dominant Conditions):
# - Solar Zenith Angle is 'Low'
# - Clear Sky Index is 'High'
# - Cloud Cover is 'Low'
# CONSEQUENT (Key Averaged Coefficients):
# - Coeff for Clear Sky GHI: +0.85
# - Coeff for Cloud Cover: -0.15
# - Coeff for Solar Zenith Angle: -0.45
```

**Output from LLM (e.g., Gemini-Pro):**

*This rule describes ideal conditions for high solar energy generation. It applies when the sun is high in the sky (a low solar zenith angle) and the atmosphere is very clear with almost no clouds. The model's prediction in this scenario is driven most strongly by the 'Clear Sky GHI' (a measure of theoretical maximum sunlight), which has a large positive influence (+0.85). The sun's high position also contributes positively (a low zenith angle combined with a negative coefficient of -0.45 results in a positive term). Finally, the model notes that even a low amount of cloud cover still has a minor negative impact on the final prediction.*

Figure 4: Case study of LLM-powered verbalization. The structured L-FMLC meta-rule is provided as a prompt, which the LLM synthesizes into a fluid natural language explanation.

