# OpenReview forum: "L-FMLC: End-to-End Neuro-Fuzzy Learning for Adaptive and Scalable Interpretability"
_ICLR.cc/2026/Conference — Submitted to ICLR 2026_

### Official Review · Reviewer_YVJP · 2025-10-20

**Soundness:** 3
**Presentation:** 3
**Contribution:** 2
**Rating:** 2
**Confidence:** 4

**Summary:**

The authors introduce a new neurosymbolic fuzzy system, L-FMLC, that builds on their prior work, FMLC. Intuitively, L-FMLC (1) clusters each feature to create an intermediate representation, (2) applies a neural network to this representation, outputting a set of weights, and (3) uses these weights as a linear predictor of the target variable. Post-hoc, the method approximately extracts rules from a learned L-FMLC for interpretability purposes. The model is evaluated on 3 datasets, showing better performance than its predecessor and multiple competitors, but worse than some specific neural networks. The goal of the work is interpretability.

Key statement (line 88): "L-FMLC contributes a method that is intrinsically interpretable at the local level and scalable to a global, high-level understanding."

**Strengths:**

- The approach is an effective extension to the prior work, clearly improving the model's interpretability and accuracy.
- The model is *locally* interpretable in terms of the input features: each prediction is modelled as a linear function of the input features, where the weights of the function are locally predicted from the input features. Of course, this 'local prediction' of the weights also *costs* interpretability; it's partially interpretable.
- The model provides some *global* "interpretability" by extracting fuzzy logic rules in a post-hoc way (also see weaknesses).
- The model is a proven universal approximator, similar to neural networks.
- The datasets are varied in tasks and input types: tabular regression, time-series regression, tabular classification.

**Weaknesses:**

- There is no (experimental or otherwise) comparison to other neurosymbolic systems. Similarly, the field of concept-based explainable AI [1] is highly related (intrinsic interpretability) but not mentioned in the related work. Less importantly, the authors do not compare experimentally to any concept-based model (CBM) (see next weakness); I understand that an experimental comparison is non-trivial as CBMs are mostly used on images with predefined concepts, but tabular CBMs exist too, and one could use the input features as concepts (thus skipping the concept predictor).
- The proposed model is quite similar (but still different) to some existing CBMs where a neural network is used to locally parametrize the weights of an interpretable function (e.g. [2, 3] where a logic rule is parametrized, and [4, 5] where a linear layer is parametrized). This provides the same local interpretability and expressivity (they are universal classifiers). The differences and similarities should be discussed in the paper (ideally, both qualitatively and quantitatively).
- The global interpretability is post-hoc, not "intrinsically-interpretable". I think a better term would be *globally explainable* instead of *globally interpretable*.
- The global interpretability is very approximate. I count at least 3 approximations (thresholding the fuzzy membership scores (line 207), averaging the modulator vectors (line 209), clustering the rules and replacing the antecedent by the medoid's antecedent and the consequent with the average consequent (lines 215-...)). The results do measure the fidelity of the approximate explanation (the "meta-rules") using Pearson correlation, but to me, this is not easy to interpret (I cannot say how "good" the found correlation value is). It would be interesting and more intuitive to know the loss in accuracy that would result from using these meta-rules compared to using the entire model.
- Surrogate models seem quite close to the global interpretability component (see previous weakness), yet they are not discussed.
- Some necessary background information is not explained (e.g. silhouette score, line 240).
- I believe some statements are incorrect (see Questions 1-2).
- There are no limitations in the conclusion.

I am willing to significantly increase my score if these concerns are addressed.

[1] Concept-based explainable AI: a survey. Poeta et al.

[2] Interpretable neural-symbolic concept reasoning. Barbiero et al.

[3] Interpretable concept-based memory reasoning. Debot et al.

[4] V-CEM: bridging performance and intervenability in concept-based models. De Santis et al.

[5] Concept Bottleneck Models with Additional Unsupervised Concepts. Sawada et al.

**Questions:**

1. Why do you call XGBoost an interpretable model (line 85, line 281)? While a single shallow decision tree is interpretable, surely no one can interpret a big ensemble?
2. You claim you "significantly outperform all other models" on the Solar dataset, but in Table 1, the Transformer baseline performs better.
3. Why do you only apply the Overlap Regularization (line 180) to consecutive pairs in the sorted list?
4. Why is Overlap Regularization necessary if you have Coverage Regularization? Can you explain this in the text?

Typos:
- Almost every citation uses \cite when it should be \citep.

---

### Official Review · Reviewer_j1XP · 2025-10-21

**Soundness:** 3
**Presentation:** 2
**Contribution:** 2
**Rating:** 4
**Confidence:** 3

**Summary:**

L-FMLC introduces an adaptive fuzzification layer and rule distillation to enhance interpretability in neuro-fuzzy systems. It learns optimal fuzzy sets and condenses rules into meta-rules, achieving strong predictive performance while maintaining clarity—proving regularization aids both interpretability and generalization.

**Strengths:**

The paper has a clear motivation: addressing the limitations of static fuzzification and rule explosion in prior work. Its technical approach of using an adaptive layer with pruning and rule distillation is a logical and targeted response to these issues.

**Weaknesses:**

1. The entire paper appears rushed, from the graphical representations to the writing. Although the guidelines technically only specify a maximum of nine pages, this is the first time I’ve actually seen someone write just over eight pages.

2.Although I understood the methodology, I believe it is not particularly easy to comprehend. There is a lack of some legends, clearer explanations, symbolic descriptions, or a preliminaries section. This is especially problematic since the foundational work referenced in the paper, 'Fuzzy-modulated linear consequents for enhanced performance and interpretability in large models,' has not yet been published and is unavailable.

3.Many references are missing. For instance, the idea of using hypernetworks to construct interpretable models is not entirely new [1,2]. While the individual modules in the algorithm section are reasonable, many aspects remind me of previous work on differentiable rule learning, including but not limited to [3,4]. Completely omitting references to others' work is unacceptable, and clearer comparisons would help better articulate the novelty of this work.

4.There are also several issues with the experiments. For example, in Table 2, the authors claim their model is the "best interpretable," but it actually underperforms compared to XGBoost and EBM, both of which are categorized as interpretable models in the related work. Moreover, the experimental results do not appear sufficiently competitive, especially given that only three datasets were used.

[1] Mahfoud, Mohammed, et al. "Learning decision trees as amortized structure inference." arXiv preprint arXiv:2503.06985 (2025).

[2] Yang, Yang, Wendi Ren, and Shuang Li. "Hyperlogic: Enhancing diversity and accuracy in rule learning with hypernets." Advances in Neural Information Processing Systems 37 (2024): 3564-3587.

[3] Xu, Sascha, Nils Philipp Walter, and Jilles Vreeken. "Neuro-Symbolic Rule Lists." arXiv preprint arXiv:2411.06428 (2024).

[4] Kusters, Remy, et al. "Differentiable rule induction with learned relational features." arXiv preprint arXiv:2201.06515 (2022).

**Questions:**

See Weakness

---

### Official Review · Reviewer_NQKf · 2025-11-01

**Soundness:** 3
**Presentation:** 3
**Contribution:** 3
**Rating:** 6
**Confidence:** 3

**Summary:**

This paper presents L-FMLC, an end-to-end neuro-fuzzy learning framework that aims to unify the adaptability of deep learning with the interpretability of fuzzy logic. Building on prior work (FMLC), the authors propose a learnable fuzzification layer that jointly learns the positions, shapes, and number of fuzzy sets per input dimension. A regularized gating mechanism prunes redundant fuzzy sets, guided by interpretability-based constraints (sparsity, overlap, and coverage).
The second contribution is a two-stage rule distillation process, combining architectural pruning and hierarchical clustering to condense thousands of instance-level fuzzy rules into a compact set of “meta-rules,” which can further be translated into natural language using large language models.
Empirical studies across regression and classification tasks show that L-FMLC achieves performance on par with strong baselines such as XGBoost and EBM, while providing substantially improved interpretability. Ablation experiments demonstrate that interpretability-driven regularization improves both semantic clarity and generalization performance.

**Strengths:**

1. Novel framework: The learnable fuzzification layer with differentiable gating is a clear conceptual and technical advancement.
2. Comprehensive interpretability pipeline: From fuzzy partitions to language-level explanations, the framework provides a full chain of transparency.
3. Robust regularization design: The three interpretability-driven losses (sparsity, overlap, coverage) are intuitive and empirically justified.

**Weaknesses:**

1. Scalability and computational complexity: The clustering-based rule distillation may not scale well to very large datasets; this limitation is not thoroughly analyzed.
2. Sensitivity to hyperparameters: Regularization weights (λ_sp,λ_ov,λ_cov) are crucial but tuned manually; an adaptive or data-driven strategy would strengthen robustness.
3. Limited comparison with other interpretable deep models: The experiments mainly contrast with tree- and rule-based baselines (XGBoost, EBM); comparisons with recent interpretable neural models (e.g., prototype-based or concept bottleneck networks) would be insightful.

**Questions:**

The paper introduces three interpretability-driven regularization terms controlled by λ_sp, λ_ov, and λ_cov. Could the authors elaborate on how sensitive the model performance and fuzzy partition structure are to these hyperparameters? In particular, is there any principled or empirical guideline for selecting their values across different tasks?
Regarding the natural language explanations generated by the LLM, what evaluation methods were applied—did the authors conduct human assessments or automatic metrics to verify the consistency between the generated explanations and the original fuzzy rules?

---

### Meta-Review · Area_Chair_aSa7 · 2025-12-30

**Summary:**

This paper introduces a fuzzy layer on top of a neural backbone to increase interpretability and flexibility of neural predictor models. The proposed architecture falls within the paradigm of (neuro-symbolic) concept-based models CBMs (despite not being presented in this way), and is a follow-up of another recent work by the authors (*). This architecture is tested on a number of predictive tasks and claimed to be more interpretable and accurate than other baselines such as xgboost.

Some reviewers are, righfully imho, skeptical of the proposed neuro-fuzzy architecture and highlight how there is no discussion nor empirical comparison to other NeSy systems and CBMs (YVJP, NQKf). Other reviewers criticize the lack of details and the poor presentation (the paper feels rushed or incomplete, j1XP, YVJP). There are overclaims in terms of the enhanced interpretability (YVJP) especially when compared to classically-interpretable baselines (j1XP). The most positive reviewer (NQKf) asked about a sensitivity ablation. Unfortunately, the authors never engaged in the rebuttal.



(*) the citation to FMLC, the previous work, despite being anonymized, is not helping the paper. It would have been better if the authors would have simply introduced the FMLC architecture fully (or provided a workable reference) and without stating it was their previous work.

**Reviewer Concerns:**

There was no rebuttal phase.

**Reviewer Scores:**

There was no rebuttal phase.

---

### Decision · Program_Chairs · 2026-01-26

Reject